# Evaluation of Meat Quality in Duhu Hybrid Lambs Reared in Different Conditions

**DOI:** 10.3390/foods13233969

**Published:** 2024-12-09

**Authors:** Wanhang Jia, Jiaxin Yang, Binglei Zhang, Saiyi Sun, Xueru Dou, Guoyan Ren, Yuqin Wang

**Affiliations:** 1College of Animal Science and Technology, Henan University of Science and Technology, Luoyang 471023, China; jwh980323@163.com (W.J.); yang18317549549@163.com (J.Y.); z18790606037@163.com (B.Z.); sunsaiyi2023@163.com (S.S.); 13343795208@163.com (X.D.); 2College of Food & Bioengineering, Henan University of Science and Technology, Luoyang 471023, China; renguoyan@163.com

**Keywords:** feeding methods, meat quality, volatile flavor compounds, amino acids, fatty acids

## Abstract

In the western Henan agricultural area, Duhu (Dupo♂ × Hu sheep♀) hybrid lambs are the primary breed of local meat sheep, predominantly raised in large-scale indoor feeding systems, although many farmers still rely on grazing. However, limited research exists on the meat quality of Duhu lambs under both grazing and indoor feeding systems. This study examined how grazing and indoor feeding affect the nutritional quality, flavor, amino acid profile, and fatty acid composition of 7-month-old Duhu lamb meat. Grazed lamb meat exhibited significantly (*p* < 0.05) higher moisture, protein content, hardness, adhesiveness, springiness, rubberiness, chewiness, and resilience than indoor-fed lamb. Regarding aroma, ammonia, oxidized compounds, and inorganic sulfides were more pronounced and stable in grazed lamb meat. Flavor analysis showed stronger bitter, salty, and sweet profiles in grazed lamb meat, whereas the sour flavor was more pronounced in indoor-fed meat. Among the volatile flavor compounds, 26 organic compounds were identified in grazed lamb meat compared with 12 in indoor-fed meat, with 1 compound common. Additionally, 16 amino acids were found in both feeding systems, with amino acid levels significantly higher (*p* < 0.01) in indoor-fed lamb. In total, 25 fatty acids were detected in grazed lamb meat, whereas 15 were found in indoor-fed meat, with 11 showing significantly different levels (*p* < 0.05). Indoor-fed lamb meat exhibited a considerably higher saturated fatty acid content (*p* < 0.05) compared to grazed lamb meat, while the n-3 polyunsaturated fatty acid content was significantly lower (*p* < 0.05).

## 1. Introduction

Mutton is a highly nutritious meat known for its high protein, high-fat, and rich trace element and mineral content in China [1]. It also aligns well with the modern concept of a low-fat, healthy diet, especially as China’s economy grows and consumers’ living standards improve [2]. With the rising demand for mutton, flavor has become a key quality indicator for consumers [3]. Meat flavor, including taste and aroma, is generated by chemical reactions between water-soluble precursors and lipids in the muscle [4,5].

Grazing and indoor feeding, the two main modes of mutton production, yield meat with different qualities and flavors due to the various feeding environments [6,7,8]. Compared with grazing, indoor-fed sheep exhibit faster growth, a higher carcass quality, and more tender meat [9,10,11]. However, grazing produces mutton with less odor (as mutton odor), more flavor, and a higher n-3 fatty acid content, which benefits human health [12,13,14,15].

The Duhu hybrid sheep, a cross between Dubo and Hu sheep (Dupo♂ × Hu sheep♀), is known for its strong reproductive ability, short breeding cycle, and high reproductive efficiency, potentially producing lambs twice a year [16]. These traits help increase sheep populations rapidly. Duhu sheep grow quickly, produce high-quality, firm, and juicy meat, and resist many common diseases, making them well-suited for grazing and reducing breeding costs. In western Henan, most farmers use Duhu sheep as terminal sires to crossbreed with Dubo sheep, capitalizing on hybrid vigor to produce high-quality commercial mutton and improve breeding efficiency.

Indoor feeding and grazing are the two primary methods of sheep farming, contributing substantially to producing high-quality mutton for the market. In China, grazing and breeding are concentrated in the north, and grazing is prevalent in the south. In the central region, especially in western Henan’s agricultural and forestry staggered areas, large-scale grazing dominates goat husbandry, supplemented by smaller-scale grazing operations. Grazing is a crucial economic activity for farmers, and the Duhu hybrid lamb is the primary breed used for meat production. However, few studies have examined the quality of mutton from grazed sheep in this area, and research specifically focused on grazing Duhu lambs in western Henan’s agricultural region is lacking.

Although research in China has focused on germplasm resources, genetic breeding, and crossbreeding, comprehensive, systematic studies on mutton quality, especially the volatile flavor compounds in mutton from sheep raised under grazing and indoor feeding systems, remain limited. This study evaluates the dietary quality, nutritional composition, and flavor of the longissimus dorsi muscle (LDM) in Duhu lambs raised under grazing and indoor feeding conditions. This research explores the development of flavor precursors that develop into mutton flavor, identifies key quality changes in Duhu lambs under varying feeding conditions, and provides a foundation for producing high-quality mutton in agricultural regions.

## 2. Study Methodology

### 2.1. Animals and Sample Collection

The Henan University of Science and Technology’s Institutional Animal Care and Use Committee approved all animal studies (approval number: HAUSTEAW-2021-C00227). Twenty-four 4-month-old hybrid Duhu female lambs from a breeding cooperative in the Yiyang Agricultural District of Luoyang City, Henan Province, were selected and divided into grazing and indoor feeding groups. Ear tags were used to identify the lambs. The grazing group followed the main herd and was managed under a uniform schedule, with grazing from 08:00 to 12:00 and again from 15:00 to 19:00. The grazing pasture encompasses an area of approximately 100 hectares, situated at an elevation of 600 m above sea level. The pasture included over 10 forage species, including sour jujube, ramie, moss, peach trees, wild rapeseed, *hispidus*, *Artemisia argyi*, and *Sophora japonicum* (Table 1). The grazing area allowed free access to food and water without supplementary feeding. Salt bricks were provided in the sheep’s house for salt supplementation. The indoor feeding group was sent to Henan Kun Yuan Farming Animal Husbandry Co. Ltd., Pingdingshan, China, where feeding and management followed the farm’s standardized method. Twelve lambs were housed in naturally ventilated pens measuring 30 m^2^, each equipped with automatic water dispensers and fed manually. Their basal diet primarily consisted of a mixed ration supplemented with fattening feed (Table 2), and the study’s experimental period extended from 15 July 2023, to 13 October 2023. The trial period lasted 90 days. Six lambs with consistent development from each group were randomly selected for slaughter at 7 months of age. They were then taken to a slaughterhouse 50 km away from the experimental site and slaughtered after 12 h of fasting with water available. The average body weight was 35.2 ± 1.0 kg for the grazing group and 40.5 ± 1.0 kg for the indoor feeding group. The Laboratory Animal Guidelines for Euthanasia (GB/T 39760-2021 [17]) recommend intravenous administration of sodium pentobarbital at a dosage of 100 mg/kg for the euthanasia of sheep in China. Following euthanasia, the carcass was dissected, and the carcass was sampled after stripping of fur and viscera. After slaughter, the LDM from the left side of the carcass was harvested, and subcutaneous fat was removed to prepare samples for index determinations.

#### Diet Composition in Grazed and Indoor-Fed Duhu Lambs

Table 1 depicts the nutritional composition of the principal forages in the pastures of the grazing group. A total of eight forages with a high feed intake by lambs were collected. Ash, crude fat, protein, ADF, NDF, DM, and starch were measured, and the results were expressed as %.

Table 2 presents the composition and nutritional value of the basal diet in the indoor feeding group, which was primarily composed of a mixed ration supplemented with fattening feed.

### 2.2. Analysis of the Physicochemical Properties of the Muscle

After slaughter, the LDM from the left carcass was used to measure water content, fat content, meat color, meat texture, shear stress, pH, and water retention. The AOAC method (2005) was used to measure and report the water, protein, and crude fat contents in the LDM muscle [18].

Three tests were performed on each sample, and the average of the results was calculated. A CM-600D colorimeter (Konica Minolta, Inc., Tokyo, Japan) was used to measure the color stability of the meat. A Magnetics PHBJ-260 portable pH meter (INESA Scientific Instrument Co., Ltd., Shanghai, China) was used to measure the pH values of the LDM muscles 24 h after mortality. The data were averaged from four randomly selected locations on each sample’s surface.

Approximately 50 g of meat samples was placed in cooking bags, the air was removed, and the bags were sealed. The water-holding capacity was then determined by calculating the weight loss percentage before and after heating them for 30 min at 71 °C in a water bath.

The samples were then cooled to room temperature, refrigerated overnight at 4 °C, and cut into 2.0 × 2.0 × 1.5 cm pieces along the muscle fiber direction. An Instron 5544 TA-XT Plus texture analyzer (Instron Corporation, Parker, MA, USA) was used to determine the results.

Another 50 g of meat samples was processed as described above, except that they were cut into 1.0 × 1.0 × 1.5 cm pieces and tested using a C-LM3B tenderness tester (Beijing Tianxiang Feiyu Technology Co., Ltd.; Beijing, China) [19].

### 2.3. Muscle Aroma and Taste Determination

Following Wang et al.’s [20] protocol, the LDM from the left carcass was subjected to an E-nose analysis, with minor adjustments, employing a portable PEN3 electronic nose (Win Muster Airsense Analytics Inc., Schwerin, Germany). Ten sensor probes such as W3S (long-chain alkanes), W2W (aromatics and organic sulfur compounds), W2S (ketones, aldehydes, and alcohols), W1W (sulfur compounds and terpenes), W1S (methane, broad range of compounds), W5C (terpenes, aromatics, and alkanes), W6S (hydrogen), W3C (ammonia and aromatic compounds), W5S (nitrogen oxides), and W1C (aromatic compounds) comprised the PEN3 system [21,22]. In short, 1 g of LDM from the left carcass was placed in a 20 mL vial headspace. The sensors absorbed the sample’s volatile gas at 400 mL/min via a hollow needle connected to a tube. There was a sixty-second detection time. The sensor was cleaned using pure air filtered by activated carbon after the sample till the sensor signals returned to normal. The sensor response values were expressed as the ratio of G to G0, where G represents the conductance of the sensor after exposure to the volatile components of the detected sample, and G0 denotes the conductance in clean air.

The method of Du et al. [23] was employed to do the E-tongue analysis of the LDM from the left body, with some small changes. An electronic tongue (Astree, Alpha MOS, Toulouse, France) with 7 sensors and 1 reference electrode (Ag/AgCl) was used to study the different tastes: the PKS (other complex tastes) and CPS (other complex tastes), as well as the AHS (sourness), CTS (saltiness), NMS (umami), ANS (sweetness), and SCS (bitterness) [24]. In short, 40 g of LDM from the left carcass was pulverized and mixed with 200 mL of distilled water at 40 °C for 60 s. The mixture was then subjected to centrifugation (Heraeus Megafuge 8R, Thermo Fisher Scientific, Waltham, MA, USA) at 8000 rpm for 15 min at 4 °C. The collected supernatant was then subjected to E-tongue analysis following the calibration and diagnosis of the sensors. The final filtered liquid was employed for the study after the data acquisition sequence in the E-tongue was switched with ultra-pure water. Each filter’s data acquisition time was 120 s. AlphaSoft v. 16 software was used to transform the E-tongue detection results to taste values (Alpha MOS, Toulouse, France).

### 2.4. Determination and Analysis of Volatile Flavor Compounds in the Muscle

According to Xu et al., the LDM from the left carcass was subjected to the HS-SPME to extract volatile compounds [25]. In short, 1.68 µg/μL 2-methyl-3-heptanone (1.5 µL, internal standard) was introduced to a 20 mL headspace vial containing 2 g of ground roasted lamb (Hamai Instrument Technology Co., Ltd., Ningbo, China). For 20 min, the equilibration of the headspace vial was carried out at 50 °C. The SPME fiber (DVB/PDMS extraction head, 65 μm, Supelco, Bellefonte, PA, USA) was left in the vial’s headspace for 40 min at 50 °C. Lastly, the SPME fiber’s extraction head was moved into a GC inlet and desorbed for two minutes at 200 °C.

A GC–MS system (QP2010 Shimadzu, Kyoto, Japan) equipped with a DB-WAX column (30 m × 0.25 mm × 0.25 μm) was used. The initial oven temperature of the gas chromatography (GC) was set at 40 °C for 3 min, subsequently raised to 120 °C at 5 °C per minute rate, and finally elevated to 200 °C at a rate of 10 °C per minute, with a hold duration of 13 min. The ion source temperature was 200 °C. The mass spectrometer was fully scanned across a mass-to-charge ratio range of 35 to 500. Volatile compounds were identified qualitatively by comparing their linear retention index (LRI) with authentic compounds and searching in the mass spectrometry database (NIST). The concentrations of the compounds were determined by dividing their peak areas by the internal standard (2-methyl-3-heptanone) peak area and multiplying the obtained value by the internal standard initial concentration (expressed in ng/g).

### 2.5. Muscle Amino Acid and Fatty Acid Determination Analyses

The amino acid content of the LDM was measured using a Sykam S-433D automatic amino acid analyzer (Sykam Scientific Instruments Co., Ltd., Beijing, China) with certain modifications [26]. About 0.2 g of dried meat samples were accurately weighed and hydrolyzed in 10 mL of 6 mol/L HCl at 110 °C for 23 h, with intermittent shaking for 20 min. The hydrolysate was diluted with deionized water to achieve a final volume of 100 mL. A 1 mL aliquot of the diluted solution, further diluted 200-fold, was filtered through a 0.22 µm membrane, and the amino acid content was analyzed using a Sykam S-433D automatic amino acid analyzer.

The lipid content of the LDM from the left carcass was extracted, as per the previously documented method [27], with some modifications. An Agilent 7890B-5977B gas chromatograph–mass spectrometer (Agilent, Palo Alto, CA, USA) was used to analyze the fatty acids in LDM muscles after they had been isolated and methylated in a single process. The internal standard was C19:0 (25 µL of 10.337 mg/mL), and the column specifications and field parameters were DB-5 (Agilent, Palo Alto, CA, USA) (30 m × 0.25 mm × 0.25 µm). The microwave stages consisted of 5 min at 250 W, 5 at 630 W, and 20 at 500 W, followed by a 15 min rest time. Helium was employed as the carrier gas, with a 2 mL/min flow rate. The samples were injected using an automated split/splitless injector at 270 °C, with a 1/20 split ratio and 1 μL injection volume. The column temperature was initially established at 70 °C for 5 min, after that, elevated at a rate of 25 °C/min to 200 °C, and further increased at 2 °C/min until reaching 240 °C, where it was maintained for 10 min. The interface and ion source temperatures were established at 280 °C and 230 °C, respectively, while the quadrupole temperature was sustained at 150 °C. Data were collected in full-scan mode, evaluated with the 7890B-5977B Agilent Data program, and identified using the NIST database.

### 2.6. Data Processing and Analysis

Excel 2021 was used for data processing. SPSS (v.22.0) software (IBM Corporation, Chicago, IL, USA) was used to analyze all data using Tukey’s test and one-way ANOVA to identify significant differences with *p* < 0.05. Two-tailed t-tests were used to evaluate the differences, with a significance level of *p* < 0.05. All values are shown as means ± standard deviations. OriginPro 2022 software (Origin Lab, Hampton, MA, USA) was used to perform mapping analysis, including principal component and correlation analyses. The findings are displayed in correlation charts.

## 3. Results

### 3.1. Analysis of the Muscle’s Physicochemical Properties

The water content, proteins, crude fat, texture characteristics (such as hardness, adhesiveness, cohesiveness, gumminess, chewiness, and resilience), and a* (colorimetric properties) of the LDMs of grazed and indoor-fed Duhu lambs were found to differ significantly (*p* < 0.05) (Table 3). No substantial variations (*p* > 0.05) were observed in shear force, water-holding capacity, pH, springiness (texture), and L and b* (colorimetric value).

### 3.2. Muscle Aroma and Taste Determination Analyses

#### 3.2.1. Comparison of LDM Aroma Between Grazed and Indoor-Fed Duhu Lambs

As shown in Figure 1, electronic nose technology was used to analyze the principal aroma components of LDMs in grazed and indoor-fed Duhu lambs. The results indicate that the meat flavors of these lamb groups can be distinguished. The results suggest that the W5S and W1W sensor values for the LDMs of indoor-fed Duhu lambs were higher than those for grazing lambs, indicating that they had higher contents of nitrogen oxides, terpenes, aromatics, and organic sulfur. Figure 2 shows a radar map of the odor values corresponding to 10 sensors. In contrast, other sensor values showed no significant difference.

#### 3.2.2. Comparison of LDM Taste Between Grazed and Indoor-Fed Duhu Lambs

Principal component analysis of the taste profiles of LDMs from grazed and indoor-fed Duhu lambs was conducted using electronic tongue technology (Figure 3). Results showed that the meat taste of these two lamb groups could not be fully distinguished, suggesting a strong similarity in taste between meat from the two feeding methods. Figure 4 presents a radar chart analysis, which shows that the response values for AHS and CPS in grazed lambs were significantly higher than those in indoor-fed lambs, indicating that sourness was the predominant taste. On the other hand, indoor-fed lambs had significantly greater reaction values for ANS, SCS, and CTS than grazed lambs, demonstrating that the major tastes were sweetness, bitterness, and salty. The PKS and NMS response values for the two feeding techniques did not differ significantly.

### 3.3. Identification and Evaluation of the Muscle’s Volatile Flavoring Compounds

According to the GC–MS results, 26 and 12 volatile organic compounds were detected in the LDMs of grazing and indoor-fed Duhu lambs, respectively, with 1 compound common to both groups (Table 4). The organic compounds identified in the LDMs of grazed lambs were classified as follows: six hydrocarbons, one alcohol, two ketones, six esters, four acids, and seven other compounds. No aldehydes were detected. In contrast, the organic compounds in the LDMs of indoor-fed hybrid lambs included two aldehydes, one hydrocarbon, two alcohols, one ketone, two esters, zero acids, and four other compounds. These findings showed that, in terms of type and amount, the volatile taste compounds were more prevalent in the LDMs of grazed lambs than in lambs fed indoors.

### 3.4. Muscle Amino Acid and Fatty Acid Determination Analysis

#### 3.4.1. Amino Acid Composition Analysis of LDMs from Grazed and Indoor-Fed Duhu Lambs

Table 5 displays the amino acid content found in the LDMs of Duhu lambs that were fed inside and on pasture. The LDMs of these lamb groups had a total of 16 amino acids. A significant difference (*p* < 0.01) was found between the LDMs of Duhu lambs reared under indoor feeding environments and those raised under grazing conditions regarding the amino acid content. In the LDMs of grazed and indoor-fed Duhu lambs, sweet amino acid content was 32.93% and 32.53%, bitter amino acid content was 39.34% and 39.94%, umami amino acid content was 27.74% and 27.53%, essential amino acid content was 39.88% and 40.25%, respectively. According to Table 6, the percentage of non-essential amino acids was 60.12% and 59.75%, respectively. Bitter, umami, essential, and non-essential amino acid contents did not significantly differ between the two groups, in contrast to the sweet amino acid level.

#### 3.4.2. Fatty Acid Composition Analysis of LDMs from Grazed and Indoor-Fed Duhu Lambs

Table 7 shows 25 and 15 fatty acids were detected in the LDMs of grazed and indoor-fed Duhu lambs, respectively. The fatty acid profile of grazed lamb meat was more varied, since all 15 fatty acids found in the LDMs of indoor-fed lambs were also found in grazed lambs. Among the shared fatty acids, C10:0, C12:0, C14:0, C14:1, C16:0, C16:1, C17:0, C18:0, C18:1n-9c, C18:2n-6c, and C20:4n-6 exhibited significant differences (*p* < 0.05). C14:0, C14:1, C16:0, C16:1, C17:0, C18:0, C18:1n-9c, and C20:4n-6 were also significantly different. The proportions of various fatty acids are shown in Table 8. Saturated fatty acids accounted for 47.28% in grazed lambs and 52.57% in indoor-fed lambs. Monounsaturated fatty acids accounted for 32.55% and 38.31% in grazed and indoor-fed lambs, respectively, whereas polyunsaturated fatty acids accounted for 20.17% and 9.11%, respectively.

## 4. Discussion

Meat’s nutritional qualities, including its moisture content, protein content, and crude fat content, significantly impact customer acceptance and market consumption potential [3,28,29]. Our analysis revealed that the longissimus muscle of grazed Duhu lambs showed higher protein and lower crude fat content than indoor-fed Duhu lambs (Table 3). This discrepancy might result from higher energy metabolism and muscle activity, which promote larger myoprotein production and higher protein content while decreasing fat deposition [30,31]. Animals raised inside showed a similar pattern, with increased glucose, lactate, and propionate absorption promoting fat deposition. Furthermore, flavor intensity and fatty acid composition correlate with crude fat level, both of which affect human health [32]. High crude fat content in the longissimus muscle is deemed unhealthy due to its association with excessive fat intake, which is linked to obesity and coronary heart disease.

Meat color, defined by attributes such as *L**, *a**, and *b**, also affects consumer perception of meat quality [33,34,35]. Compared to indoor-fed Duhu lambs, grazed Duhu lambs demonstrated lower *a** values and higher *b** values; consequently, the color attributes of the meat from indoor-fed Duhu lambs were more appealing (Table 3). Because meat color can be influenced by fat deposition and oxidation [36], the lower *L** value in pasture-grazed lambs may reflect their lower fat content and higher metabolic oxidation due to increased physical activity [37]. The reduced *b** values in indoor-fed lambs may be associated with their increased crude fat content [38]. Similarly, Ke et al. found that lambs raised on pasture had lower *L** values and similar or lower *a** and *b** values than lambs raised indoors [39]. Meat discoloration may also result from reduced myoglobin oxidation caused by lower phenolic compounds in the concentrated diets of animals raised indoors [33] (Table 1 and Table 2).

Meat tenderness and juiciness are vital to consumer acceptance and satisfaction [40,41]. Studies comparing tenderness and juiciness in pasture-grazed and indoor-fed lambs found pasture-grazed animal meat less tender but comparably juicy than their indoor-fed counterparts [31]. Lower fiber content, weaker connective tissue, and the breakdown of myofibrillar proteins and connective tissue may all contribute to the increased softness of Duhu lamb muscle raised indoors [42]. Maiorano et al. proposed that the increased average daily gain in lambs correlates with higher soluble collagen content, which may improve tenderness [43]. The results demonstrate that the longissimus muscle of grazed Duhu lambs demonstrated lower tenderness and juiciness than indoor-fed lambs (Table 3). In particular, hardness, adhesiveness, cohesiveness, gumminess, chewiness, and resilience were all higher in grazed Duhu lambs than in store-fed Duhu lambs, likely due to their increased physical activity in the pasture, resulting in more compact muscles and dietary differences (Table 1 and Table 2).

Flavor is a crucial factor in the eating quality of mutton, affecting consumers’ long-term purchase decisions. In the present study, the W5S and W1W sensor responses were significantly lower for grazing mutton than for indoor-fed mutton, reflecting the latter’s stronger W5S and W1W responses and suggesting that indoor-fed mutton contains more inorganic sulfide and ammonia oxidation compounds combined with Table 4, which is consistent with the amount of VOCs detected. Regarding taste, grazed mutton had a higher response value for sourness (sour aftertaste), indicating a longer-lasting sour flavor. Due to more sour organic compounds, we detected significant amounts of acid-class VOCs in grazing mutton (Table 4). Indoor-fed mutton demonstrated an increased response to sweetness and umami, attributed to sweet and umami amino acids. The taste profile of indoor-fed mutton was primarily influenced by free amino acids and 5′-nucleotides, with the umami flavor resulting from the synergistic interaction between these components [44]. Glutamic acid was the primary contributor to umami in indoor-fed mutton (Table 5).

Although the electronic nose and tongue can provide the basic outline of smell and taste in meat, they cannot identify specific volatile compounds. Our study found that although the types of VOCs in indoor-fed mutton were small, the content of VOCs in indoor-fed mutton was high. Alcohols and aldehydes had the highest levels of volatile organic compounds (VOCs) in mutton fed indoors. Essential flavorings known as aldehydes are created when fatty acids oxidize, and amino acids undergo the Strecker reaction [45]. One of the primary sources of meat flavor is synthetic compounds and foods containing aldehydes that taste like citrus, grass, and lipids [46]. The primary source of alcohol in mutton is the oxidation and breakdown of unsaturated fatty acids. Straight-chain and branched-chain alcohols are the two categories of alcohols. Lipid oxidation is the primary source of straight-chain alcohols [47]. The Strecker degradation of amino acids and the reduction of branched-chain aldehydes are the primary processes that result in branched-chain alcohols. Low-carbon chains usually appear in these alcohols. The alcohol flavor changed from an anesthetic odor to a fruity and lipid-based aroma as the number of carbon chains increased. This change was also very consistent with the tendencies of E-tongue and E-nose.

Our analysis of volatile flavor compounds showed a greater diversity of these compounds in grazed Duhu lambs, which likely contributes to their more distinct aroma and taste. This variation is primarily due to dietary differences; grazing lambs have access to various fragrant grasses and Chinese herbs, whereas the diet of indoor-fed lambs is consistent and controlled [48] (Table 1 and Table 2).

Amino acids, the building blocks of proteins, are indicators of meat quality and protein content [48,49]. The amino acid profile primarily determines the nutritional value of meat [50]. This research indicated that the amino acid content in indoor-fed Duhu lambs exceeded that of grazed Duhu lambs. A diet with a higher concentration in store-fed Duhu lambs may enhance amino acid content relative to grazed Duhu lambs (Table 1).

The predominant amino acids identified were Glu, Asp, Lys, Leu, Arg, and Ala, which closely correspond to the amino acid profile of the provided feed. The amino acid profile aligns with earlier findings documented by Gilka et al. [51]. The amino acid content of meat determines its flavor and nutritional properties, with high levels of essential amino acids (EEAs) meeting human dietary needs [52,53,54]. The Food and Agriculture Organization of the United Nations and the World Health Organization indicate that around 40% of essential amino acids (EAAs) relative to total amino acids (TAAs) and more than 60% of EAAs compared to non-essential amino acids (NEAAs) serve as indicators of high-quality proteins [5]. In the current study, the EAA/TAA ratios for grazed and indoor-fed mutton were 39.88% and 40.25%, respectively. In contrast, the EAA/NEAA ratios were 66.35% and 67.37%, respectively, confirming that both mutton types are high-quality protein sources.

Fatty acid content plays a significant role in lamb’s flavor, nutritional value, and impact on human health [55,56]. Health professionals worldwide recommend reducing saturated fatty acid (SFA) intake owing to its strong positive correlation with cardiovascular disease [53,57,58]. The fatty acid composition of lamb meat can be changed by meal types, since different feeding systems have varied dietary fatty acid compositions (from concentrate, grass, or a mixture). In the present study, the SFA content in indoor-fed mutton was significantly higher than in grazed mutton, indicating that the latter is healthier for consumers. Stearic acid (C18:0) is a crucial component in the mutton’s odor, with its levels positively correlated with the intensity of this odor. The C18:0 content was significantly higher in indoor-fed mutton, possibly due to its stronger odor than in grazed mutton. For monounsaturated fatty acids (MUFAs), the C18:1n9c content was substantially higher in indoor-fed mutton. C18:1n9c is known to reduce cholesterol levels, making it beneficial for human health. Polyunsaturated fatty acids (PUFAs) also have health benefits. Higher PUFA levels improve meat flavor and slow lipid oxidation [5,55]. In the present study, indoor-fed mutton had higher C18:2n6c and lower C18:3n3 content than grazed mutton. C18:3n3 and C18:2n6c are essential fatty acids that must be obtained through diet. They provide neuroprotective and antiaging benefits, with C18:3n3 offering various health advantages, such as preventing cardiovascular disease and reducing blood lipid levels. Studies have shown that eicosapentaenoic and docosahexaenoic acids can help reduce hypertension and improve vascular endothelial function, supporting cardiovascular health [56,59,60]. The human body cannot synthesize n-3 PUFAs, so adequate dietary intake is essential for overall health [12,15,61]. The effects of rearing systems on PUFAs are substantial, as pasture- and forage-based diets are common sources of *n*-3 PUFAs in many livestock production systems. In this study, the n-3 PUFA content in grazed mutton was substantially higher than in indoor-fed mutton, indicating that the former is more beneficial for human health. The composition and content of MUFAs in mutton largely depend on the animal’s diet. The ratio of SFA to PUFA and n-6 PUFA to n-3 PUFA determines the nutritional value of meat. The n-6/n-3 ratio should be ≤4:1, and the PUFA/SFA ratio should be ≥0.4 for the best health advantages [56]. In the current study, the PUFA/SFA ratio of grazed mutton was 0.42, significantly greater than that of mutton fed indoors (0.17). The acceptable range was exceeded by the n-6/n-3 ratios of 5:1 in grazed and 32:1 in indoor-fed mutton. Although indoor-fed mutton’s overall fatty acid content was higher, grazed mutton is considered more nutritious and healthier based on its fatty acid profile.

## 5. Conclusions

The present study examined the impact of grazing and indoor feeding on the nutritional quality, flavor, amino acid profile, and fatty acid composition of mutton from Duhu hybrid lambs. Substantial disparities were observed in the moisture, crude fat content, and morphological properties of mutton, comparing the two husbandry techniques. Based on analyses involving electronic nose and electronic tongue technology combined with volatile flavor substances, it was found that grazed mutton has more diverse flavor substances and a richer flavor. The amino acid ratios of both grazed and indoor-fed mutton indicated their high quality, although the amino acid content was lower in grazed mutton. Notably, n-3 PUFA content was markedly higher in grazed mutton, making it a healthier option for consumers. This study also has certain limitations. First, the pH of meat samples was not measured at 45 min post-slaughter, which markedly impacted the analysis of meat quality. Additionally, the proportions of volatile flavor compounds in muscle tissue and their respective residence times were not thoroughly examined. Finally, all indicators in this study were derived from the LDM from the left side of the lamb carcass, which may limit the completeness of the experimental data. The results offer insights into the differences in mutton quality between grazing and indoor-fed Duhu hybrid lambs, potentially guiding the production of high-quality mutton meat.

## Figures and Tables

**Figure 1 foods-13-03969-f001:**
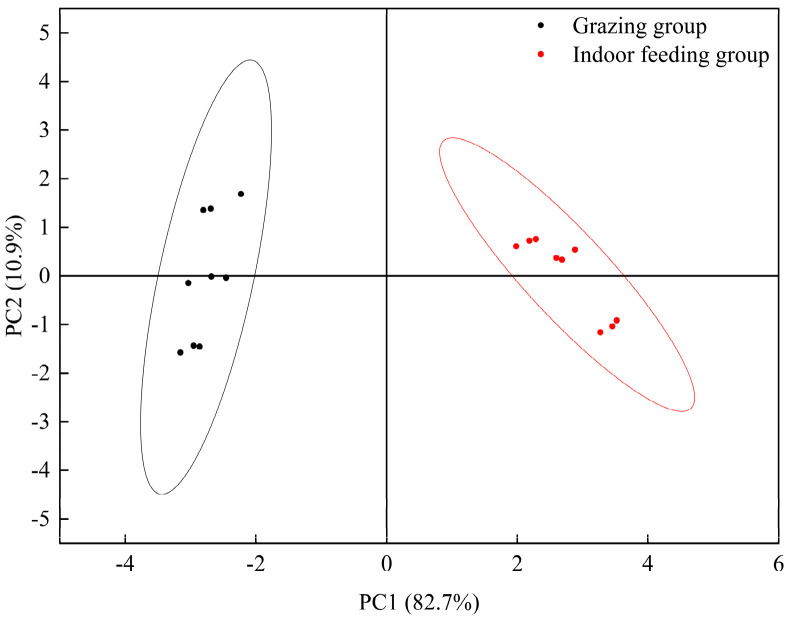
Principal component analysis for muscle electronic nose data.

**Figure 2 foods-13-03969-f002:**
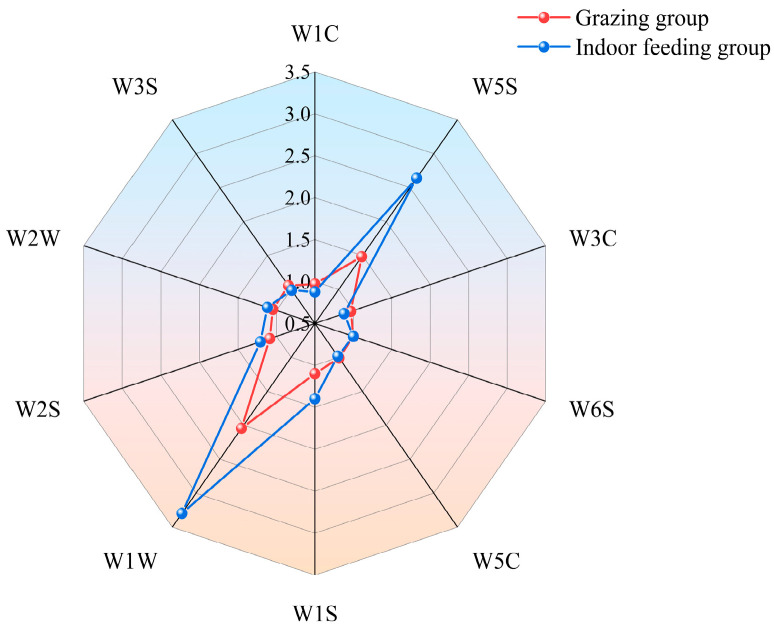
Muscle electronic tongue radar data. W3S: long-chain alkanes; W2W: aromatics and organic sulfur compounds; W2S: alcohols, aldehydes, and ketones; W1W: sulfur compounds and terpenes; W1S: methane (broad range of compounds); W5C: alkanes and aromatics; W6S: hydrogen; W3C: ammonia and aromatic compounds W5S: nitrogen oxides; W1C: aromatic compounds.

**Figure 3 foods-13-03969-f003:**
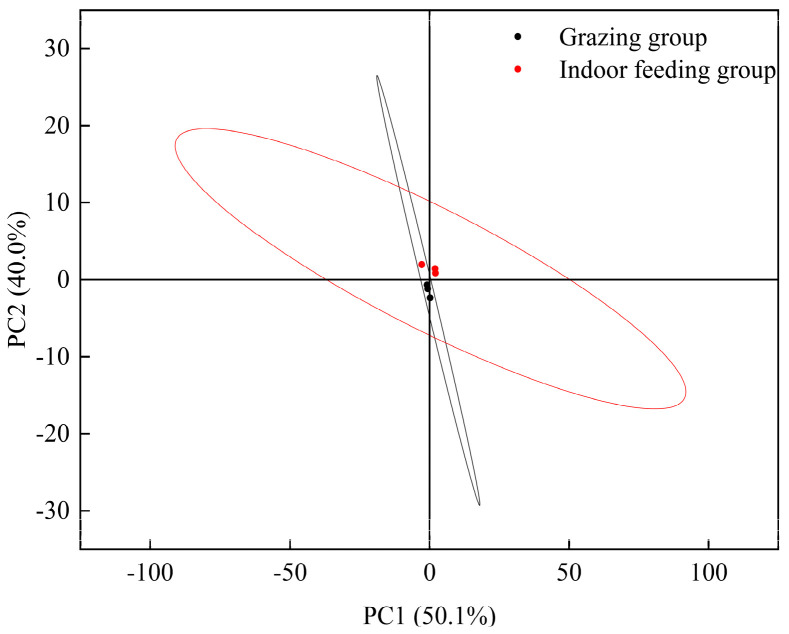
Principal component analysis for muscle electronic tongue data.

**Figure 4 foods-13-03969-f004:**
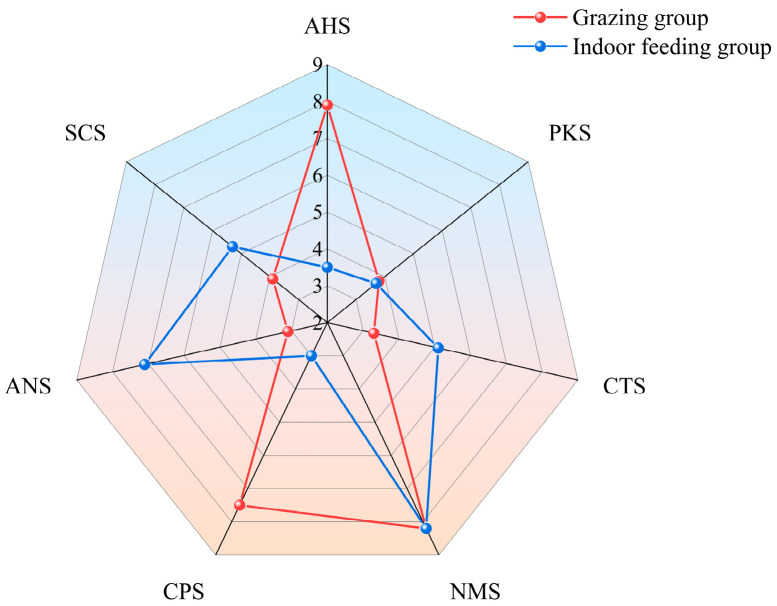
Muscle electronic nose radar data. SCS: bitterness; ANS: sweetness; NMS: umami; CTS: saltiness; AHS: sourness; PKS and CPS represent complex taste.

**Table 1 foods-13-03969-t001:** Nutritional composition analysis of primary forage species in the grazing group.

Forage	Ash/%	Crude Fat/%	Proteins/%	ADF/%	NDF/%	DM/%	Amylum/%
Wild jujube	12.55	2.60	19.70	24.63	40.08	25.91	13.84
Ramie	12.51	2.28	22.18	29.57	47.06	24.77	14.68
Stinging moss	12.61	2.30	16.67	26.80	45.82	31.52	15.53
*Prunus persica*	10.35	2.80	19.88	25.67	29.51	26.97	16.86
Wild rapeseed	12.38	2.30	21.88	29.88	49.51	18.39	13.01
*Arthraxon hispid*	15.77	2.94	21.32	34.77	53.93	17.59	14.11
*Artemisia argyi*	9.28	4.35	22.93	31.37	42.01	16.48	13.02
*Sophora japonica*	8.59	2.98	20.04	24.55	36.01	21.41	20.77

ADF: acid detergent fiber; NDF: neutral detergent fiber; DM: dry matter.

**Table 2 foods-13-03969-t002:** Composition and nutrient levels of the basal diet in the indoor feeding group.

Ingredients		Nutritional Components (%)	
Corn	63.00	Dry matter	79.79
Soybean meal	16.00	Moisture	13.5
Wheat bran	8.00	Crude ash	4.51
Hsin cathepsin	3.00	Crude protein	8.83
Baking soda	1.50	Ether extract	2.69
MgO	0.80	NDF	16.39
NaCl	1.20	ADF	6.85
Plant oil	0.80	Crude fiber	1.86
Mold inhibitor	0.30	Starch	49.91
Amino acid modulator YSS-3	0.40	ADL	1.28
5% lamb robustness premix	5.00	Calcium	1.97
Total	100.00		

The premix contained Cu (10 mg), Zn (50 mg), Fe (60 mg), I (0.6 mg), Mn (44 mg), Se (0.25 mg), Co (0.21 mg), VA (2800 IU), and VE (100 IU). Nutrient levels were measured values.; ADL: acid detergent lignin.

**Table 3 foods-13-03969-t003:** Comparison of the physicochemical properties of the longissimus dorsi muscle in grazed and indoor-fed Duhu lambs.

Projects		Grazing Group	Indoor Feeding Group	*p* Value
Water content (%)		78.69 ± 0.47 a	75.27 ± 0.5 b	0.001
Proteins (%)		23.3 ± 0.08 a	19.77 ± 0.38 b	0.037
Shear force (kgf)		6.12 ± 0.75	6.36 ± 0.27	0.656
Crude fat (%)		1.27 ± 0.23 a	1.76 ± 0.17 b	0.046
Water-holding capacity (%)		41.46 ± 1.06	38.24 ± 2.02	0.091
pH_24h_		6.05 ± 0.39	5.83 ± 0.12	0.402
Textural properties	Hardness	16113.9 ± 1868.66 a	8755.16 ± 926.19 b	0.004
	Adhesiveness	−5.51 ± 4.09 a	−1.42 ± 0.84 b	0.022
	Springiness	0.53 ± 0.01	0.48 ± 0.04	0.101
	Cohesiveness	0.6 ± 0.01 a	0.49 ± 0.01 b	0.001
	Gumminess	9591.86 ± 953.95 a	4294.32 ± 459.24 b	0.001
	Chewiness	5109.73 ± 616.72 a	2036.3 ± 20.11 b	0.001
	Resilience	0.29 ± 0.01 a	0.21 ± 0.01 b	0.001
Chromacity	L	35.21 ± 2.77	40.44 ± 3.48	0.115
	a*	11.44 ± 0.8 a	15.5 ± 0.43 b	0.004
	b*	8.22 ± 0.76	5.5 ± 1.87	0.079

Values represented by separate lowercase letters within the same row demonstrate significantly different (*p* < 0.05) physicochemical properties.

**Table 4 foods-13-03969-t004:** Muscle volatile flavor compounds in grazed and indoor-fed lamb meat.

Taxonomic	Feeding Model					
	Grazing Group			House Feeding Group		
	VOC	RT	Composition (ng/g)	VOC	RT	Composition (ng/g)
Aldehydes	ND			Hexanal	29.41	58.018
	ND			Nonanal	34.25	61.782
Hydrocarbons	2,2,7,7-Tetramethyloctane	40.707	10.372	ND		
	Tetradecane	37.507	394.776	ND		
	1-Hexyl-2-nitro cyclohexane	42.289	1.153	2,2′-Bipyrrolidine	20.176	9.674
	1-Nonylcycloheptane	39.465	3.288	ND		
	Dodecane, 1-fluoro-	38.408	11.578	ND		
	Pentadecane	39.252	248.874	ND		
Ketones	Valeric anhydride	21.816	6.857	ND		
	2,10-Dodecadien-1-ol, 3,7,11-trimethyl-, (Z)-	39.95	13.124	1-Penten-3-one	30.34	28.484
Alcohols	ND			Isopropyl Alcohol	12.72	121.182
	trans-(2-Ethylcyclopentyl)methanol	38.888	4.895	1-Hexen-3-ol	21.394	505.889
Esters	(E)-But-2-en-1-yl 2-methylbutanoate	38.455	3.822	ND		
	Carbonic acid, decyl undecyl ester	39.701	24.022	ND		
	2,2,4-Trimethyl-1,3-pentanediol di-isobutyrate	40.794	45.24	Methane, isocyanato-	30.363	26.48
	2,4,4-Trimethyl-1-pentanol, chlorodifluoroacetate	36.901	3.166	4-Methyl-6-hepten-4-olide	29.318	170.264
	Carbonic acid, eicosyl vinyl ester	40.363	24.022	ND		
	Oxalic acid, cyclobutyl octadecyl ester	38.466	7.302	ND		
Acids	Carbamodithioic acid, diethyl-, methyl ester	37.317	11.52	ND		
	3-Methyl-l-valine	4.651	90.588	ND		
	Pentanoic acid, 2-methyl cyclohexyl ester, cis-	36.745	3.464	ND		
	Succinic acid, 2-methyl pent-3-yl pentafluorophenyl ester	19.592	20.855	ND		
Other VOCs	Oxime-, methoxy-phenyl-_	16.774	35.201	ND		
	Heptane, 1,1′-oxybis-	40.654	11.392	Heptane, 1,1′-oxybis-	34.256	55.053
	Cyclohexene, 4-(1,5-dimethyl-1,4-hexadienyl)-1-methyl-	38.137	31.783	2-Acetylpiperidine	30.375	83.057
	2,2-Dimethyl-propyl 2,2-dimethyl-propane-sulfinyl sulfone	35.215	7.351	ND		
	trans-2,4-Dimethylthiane, S, S-dioxide	38.645	11.578	Oxirane, 2-(1,1-dimethyl ethyl)-3-ethyl	25.847	61.11
	Tetradecahydro-1-methyl phenanthrene	40.436	3.356	4-Butyl-1,3-thiazole	29.434	121.182
	1-Hexadecyne	40.124	14.744	ND		

VOC: volatile organic compound; RT: retention time; ND: volatile compounds not detected.

**Table 5 foods-13-03969-t005:** Amino acid content in the muscle of grazed and indoor-fed Duhu lambs.

Amino Acid Types	Grazing Group	Indoor Feeding Group	*p* Value
	Amino Acid Content (g/100 g)		
SAA			
Thr	0.72 ± 0.02 a	1.05 ± 0 b	0.001
Ser	0.63 ± 0.02 a	0.9 ± 0 b	0.002
Pro	0.51 ± 0.04 a	0.72 ± 0.01 b	0.001
Gly	0.71 ± 0.05 a	0.91 ± 0.01 b	0.002
Ala	0.91 ± 0.04 a	1.29 ± 0 b	0.003
Lys	1.36 ± 0.03 a	2 ± 0.01 b	0.001
BAA			
Val	0.75 ± 0.01 a	1.06 ± 0.01 b	0
Met	0.42 ± 0.01 a	0.63 ± 0 b	0.001
L1e	0.71 ± 0.01 a	1.03 ± 0 b	0
Leu	1.28 ± 0.03 a	1.83 ± 0 b	0.001
Tyr	0.56 ± 0.05 a	0.87 ± 0.01 b	0.007
Phe	0.62 ± 0.02 a	0.91 ± 0 b	0.001
Hs	0.46 ± 0.02 a	0.67 ± 0 b	0.002
Arg	0.98 ± 0.04 a	1.44 ± 0 b	0.002
UAA			
Asp	1.45 ± 0.03 a	2.09 ± 0 b	0.001
Glu	2.63 ± 0.14 a	3.73 ± 0.01 b	0

BAA: bitter amino acid; SAA: sweet amino acid; UAA: umami amino acid. Significant differences exist between values in the same row that have different lowercase letters (*p* < 0.05).

**Table 6 foods-13-03969-t006:** Proportions of amino acids in grazed and indoor-fed Duhu lambs.

Amino Acid Types	Grazing Group	Indoor Feeding Group	*p* Value
	Proportion of Amino Acids (%)		
SAA	32.93 ± 0.2 a	32.53 ± 0.05 b	0.026
BAA	39.34 ± 0.43	39.94 ± 0.05	0.074
UAA	27.74 ± 0.24	27.53 ± 0.01	0.21
EAA	39.88 ± 0.58	40.25 ± 0.03	0.335
NEAA	60.12 ± 0.58	59.75 ± 0.03	0.335

EAA: essential amino acid; NEAA: nonessential amino acid. Significant differences appear between values with different lowercase characters in the same row (*p* < 0.05).

**Table 7 foods-13-03969-t007:** The fatty acid composition of the longissimus dorsi muscle in Duhu lambs raised in grazing versus indoor feeding environments.

Fatty Acid Type	Grazing Group	Indoor Feeding Group	*p* Value
	Fatty Acid Composition (mg/g)		
C6:0	0.04 ± 0.01 a	0 ± 0 b	0.008
C10:0	0.01 ± 0.01 a	0.09 ± 0.03 b	0.011
C12:0	0.02 ± 0.01 a	0.22 ± 0.08 b	0.01
C14:0	0.22 ± 0.06 a	2.08 ± 0.58 b	0.005
C14:1	0.02 ± 0.03 a	0.11 ± 0.02 b	0.013
C15:0	0.05 ± 0.02 a	0.19 ± 0.06 b	0.013
C16:0	2.19 ± 0.45 a	10.92 ± 1.51 b	0.001
C16:1	0.17 ± 0.05 a	1.01 ± 0.17 b	0.001
C17:0	0.09 ± 0.03 a	0.47 ± 0.12 b	0.006
C17:1n7	0.06 ± 0.01 a	0 ± 0 b	0.011
C18:0	2.47 ± 0.55 a	8.44 ± 1.41 b	0.002
C18:1n9t	0.4 ± 0.13 a	0 ± 0 b	0.031
C18:1n9c	2.83 ± 0.61 a	15.14 ± 2.07 b	0.001
C18:2n6t	0.02 ± 0.01 a	0 ± 0 b	0.026
C18:2n6c	1.18 ± 0.19 a	2.65 ± 0.49 b	0.008
C20:0	0.02 ± 0.01 a	0 ± 0 b	0.026
C20:2	0.02 ± 0.01	0 ± 0	0.13
C22:0	0.01 ± 0.01 a	0 ± 0 a	0.158
C20:1n9	0.01 ± 0 a	0.03 ± 0.01 b	0.038
C18:3n3	0.23 ± 0.05 a	0.12 ± 0.02 b	0.023
C20:3n6	0.03 ± 0.01 a	0.07 ± 0.02 b	0.013
C20:4n6	0.44 ± 0.04 a	1.03 ± 0.12 b	0.001
C20:5n3(EPA)	0.06 ± 0.03	0 ± 0	0.093
C24:1n9	0.03 ± 0.01 a	0 ± 0 b	0.038
C22:6n3 (DHA)	0.04 ± 0.01 a	0 ± 0 b	0.006
SFA	5.11 ± 1.11 a	22.42 ± 3.77 b	0.002
MUFA	3.52 ± 0.8 a	16.29 ± 2.25 b	0.001
PUFA	2.15 ± 0.31 a	3.87 ± 0.56 b	0.017
n-3	0.33 ± 0.06 a	0.12 ± 0.02 b	0.004
n-6	1.68 ± 0.23 a	3.75 ± 0.58 b	0.005

SFAs: saturated fatty acids; PUFAs: polyunsaturated fatty acids; MUFAs: monounsaturated fatty acids. Values represented by distinct lowercase letters within the same row indicate significant differences (*p* < 0.05).

**Table 8 foods-13-03969-t008:** Proportion of fatty acids in the longissimus dorsi muscle of hybrid Duhu lambs reared under grazing and indoor feeding conditions.

Fatty Acid Type	Grazing Group	Indoor Feeding Group	*p* Value
	Proportion of Fatty Acids (%)		
SFA	47.28 ± 0.64 a	52.57 ± 0.76 b	0.001
MUFA	32.55 ± 0.89 a	38.31 ± 0.63 b	0.001
PUFA	20.17 ± 1.44 a	9.11 ± 0.45 b	0.003

Values represented by distinct lowercase letters within the same row indicate significant differences (*p* < 0.05).

## Data Availability

The original contributions presented in the study are included in the article; further inquiries can be directed to the corresponding author.

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
