# Peer review of "Evaluation of Meat Quality in Duhu Hybrid Lambs Reared in Different Conditions"

_foods, 2024, doi:10.3390/foods13233969_

Round 1
Reviewer 1 Report
Comments and Suggestions for Authors
The paper deals with the investigation of qualitative characteristics of Duhu hybrid lambs reared in two different housing systems, indoors and grazing. Physicochemical properties of the muscles of investigated lambs such as water content, shear resistance, fat content (intramuscular), water binding capacity and muscle pH were compared. In addition, the textural properties of the meat were analysed using a texture analyser. The electronic nose and tongue were used to analyse taste and odour. In addition, volatile compounds were analysed using gas chromatography.
The amino acids composition and the fatty acid profile were determined in the meat of the lambs. The differences between the groups in the measured parameters are correctly presented, but I must point out some inconsistencies in the terminology used in the paper.
In the section on materials and methods, water and fat content, meat colour and texture, pH and water holding capacity are listed as indicators of "edible and nutritional quality"," which is not correct terminology in meat science.
These parameters are usually referred to as physico-chemical properties and are not related to edibility or nutritional composition.
In the section on materials and methods, water and fat content, meat colour and texture, pH and water holding capacity are listed as indicators of "edible and nutritional quality"," which is not correct terminology in meat science. These parameters are usually referred to as physico-chemical properties and are not related to edibility or nutritional composition. The terminology should therefore be revised to avoid confusion. In the present study, it is not appropriate to discuss "edible quality"," while "nutritional quality" could be understood as fat content, amino acid and fatty acid analysis is correctly presented in the text.
In addition to displaying the pH value, please show when it was measured (45 p.m., 24 hours p.m.).
Two identical subtitles are repeated in the text:
3.3. Muscle amino acid and fatty acid determination analyses
3.4. Muscle amino acid and fatty acid determination analysis
please correct it.
Comments on the Quality of English Language
The terminology used to describe sensory properties should be refined with the help of a native English speaker. It is important to use terms such as "flavour"," "aroma" and "taste" correctly. These terms could be defined in the text so that readers can better understand the results.
Reviewer 2 Report
Comments and Suggestions for Authors
The manuscript entitled by Jia et al. "Analysis of meat quality in Duhu hybrid lambs reared in different ways" deals with the effect of rearing system (grazing vs indoor) on the quality of lamb’s meat. The work contains serious drawbacks that must be taken into account. I recommend to take into consideration the following comments:
Introduction: There are no information about the conditions of each rearing systems such as feeding, welfare, housing and husbandry practices etc. Need to be added
Lines 57-60: the text not belong here but in the material and methods
Material and methods:
Lines 65-80: Information about conditions of both rearing systems including of feeding composition, welfare, housing, then slaughtering (weight at slaughter, place of slaughtering, slaughtering process etc.) should be added.
In line 68 you mentioned that 24 animals (12 for each rearing system) were used for study, whereas at line 77-78 you indicated that ‘twelve lambs from each group with consistent growth were randomly selected for slaughter’ how can you selected 12 animals randomly with consistent growth from 12 animals?!!
Lines 78-80: you mentioned that ‘LDMs were collected and divided into two parts: one part was stored in 4% paraformaldehyde solution, whereas the other part was stored in a freezer at −80°C’ which group of samples (in 4% paraformaldehyde solution or in a freezer at −80°C) are used for analysis in this study? or both was used? Why you don’t indicate to this matter letter in your study?
Line 82: left half of carcasses?
Lines 82-83: why proteins content not determined in the meat? As we know the feeding and physical activity of animal under grazing and indoor system play important role on the protein content in the meat.
Lines 96-107, 108-118, 121-125: analysis of each of aroma and taste, Volatile flavor compounds and fatty acid were conducted according to which scientific sources? Please include the citations as a scientific source and then mentioned them in list of references
Results:
Table 1 and 2 with the same title, so what is the reason for dividing in two tables?
Line 131: you mentioned that significant differences (P < 0.05) were observed between the LDMs of grazed and indoor-fed Duhu lambs in terms colorimetric properties (a* and b*), which not in line with the results shown in the table 2.
Under all tables you mentioned that ‘Values with different lowercase letters….’ But no lowercase letters were used.
Figures 1,2,3 and 4: need to be more clear (technically).
Line 173: the title (amino acid and fatty acid) not belong the text (volatile flavor compounds), correct it please.
Table 8: was reduplicate, please correct it.
Please check the ''different lowercase letters'' and particularly in the table 8 for example for C18:1n9c (Indoor feeding group).
Discussion:
The discussion very low focus on the role of rearing system (grazing vs indoor) which is the mean factor of this study, for example the text from lines 269 till 290 is only indication to definitions and meaning of texture parameters and repeating mentioned again the results.
The discussion need to rewrite again to be more focus on the main topic ''effect of rearing system: grazing vs indoor on the meat quality''.
Conclusion:
limitations of this study can be included to the conclusion paragraph
Reviewer 3 Report
Comments and Suggestions for Authors
Analysis of meat quality in Duhu hybrid lambs reared in different ways
Review 1
General comments:
The manuscript investigates meat quality parameters of Duhu hybrid lambs reared in two different rearing systems (indoor vs. grazing). Although numerous studies have been conducted on the influence of rearing lambs indoors vs. grazing on meat quality so far, the research could potentially contribute to knowledge about the influence on the meat quality of Chinese Duhu hybrid lambs.
However, the manuscript has numerous shortcomings that must be corrected and redesigned before it can be considered for publication.
Detailed specific comments are placed directly in the attached manuscript.

Round 2
Reviewer 2 Report
Comments and Suggestions for Authors
My remained comments and suggestion was indicate in red color in pdf file

Reviewer 3 Report
Comments and Suggestions for Authors
Dear authors, I appreciate the effort you have invested in improving your manuscript, but unfortunately, it is not yet ready for publication. Despite the large amount of valuable data obtained through the research, they have not been organized and processed in the best way. The mere fact that data on aroma and flavor are presented before AA and FA profiles (as their precursors) indicates certain shortcomings. Furthermore, the statistical analysis of the data has not been explained adequately, and the presented results reveal deficiencies in this regard. It is recommended to perform additional statistical processing that will highlight the correlations between specific quality indicators, which can lead to drawing correct conclusions based on the results rather than assumptions. Please take these comments as well-intentioned and consider them seriously before reconstructing your manuscript again. Detailed specific comments are placed directly in the attached manuscript. I wish you much success in this process.
